# Rates of knee arthroplasty in patients with a history of arthroscopic chondroplasty: results from a retrospective cohort study utilising the National Hospital Episode Statistics for England

Simon G F Abram [ID] ,[1,2] Antony J R Palmer,[1,2] Andrew Judge,[1,3,4] David J Beard,[1,2] Andrew J Price [ID] [1,2]

¹Nuffield Department of Orthopaedics, Rheumatology and Musculoskeletal Sciences, University of Oxford, Oxford, UK
²NIHR Biomedical Research Centre, Oxford, UK
³Musculoskeletal Research Unit, University of Bristol, Bristol, UK
⁴NIHR Biomedical Research Centre, Bristol, UK

**Correspondence to**
Dr Simon G F Abram;
simon.abram@ndorms.ox.ac.uk

## ABSTRACT

**Objective** The purpose of this study was to analyse the rate of knee arthroplasty in the population of patients with a history of arthroscopic chondroplasty of the knee, in England, over 10 years, with comparison to general population data for patients without a history of chondroplasty.

**Design** Retrospective cohort study.

**Setting** English Hospital Episode Statistics (HES) data.

**Participants and interventions** Patients undergoing arthroscopic chondroplasty in England between 2007/2008 and 2016/2017 were identified. Patients undergoing previous arthroscopic knee surgery or simultaneous cruciate ligament reconstruction or microfracture in the same knee were excluded.

**Outcomes** Patients subsequently undergoing a knee arthroplasty in the same knee were identified and mortality-adjusted survival analysis was performed (survival without undergoing knee arthroplasty). A Cox proportional hazards model was used to identify factors associated with knee arthroplasty. Relative risk of knee arthroplasty (total or partial) in comparison to the general population was determined.

**Results** Through 2007 to 2017, 157 730 eligible chondroplasty patients were identified. Within 1 year, 5.91% (7984/135 197; 95% CI 5.78 to 6.03) underwent knee arthroplasty and 14.22% (8145/57 267; 95% CI 13.94 to 14.51) within 5 years. Patients aged over 30 years with a history of chondroplasty were 17.32 times (risk ratio; 95% CI 16.81 to 17.84) more likely to undergo arthroplasty than the general population without a history of chondroplasty.

**Conclusions** Patients with cartilage lesions of the knee, treated with arthroscopic chondroplasty, are at greater risk of subsequent knee arthroplasty than the general population and for a proportion of patients, there is insufficient benefit to prevent the need for knee arthroplasty within 1 to 5 years. These important new data will inform patients of the anticipated outcomes following this procedure. The risk in comparison to non-operative treatment remains unknown and there is an urgent need for a randomised clinical trial in this population.

### Strengths and limitations of this study

► Strengths of the data source analysed in this study include comprehensive, national, data collection and the ability to match treatment with outcomes, including by the laterality of intervention, over time.

► This is the largest cohort of patients undergoing arthroscopic chondroplasty that has been reported, with strict inclusion criteria, excluding patients with a history of previous surgery to the same knee and those undergoing simultaneous ligament reconstruction or microfracture.

► All studies of this design rely on coding accuracy and some coding errors are inevitable; although outcomes were stratified by a range of patient factors, unmeasured potential confounders include body mass index, limb alignment, baseline radiographic status.

► Knee arthroplasty is an end-stage outcome and will underestimate the true burden and severity of symptomatic osteoarthritis in this population.

► The outcome had these patients not undergone arthroscopic chondroplasty remains unknown.

## INTRODUCTION

Around 2 million knee arthroscopy procedures are performed worldwide each year.[1] Historically, knee washout and 'debridement' was shown to be ineffective for the treatment of advanced osteoarthritis.[2–4] For early osteoarthritis, however, a number of surgical and non-surgical treatments are available and treatment selection is challenging.[5] The aim of treatment in these cases is to improve symptoms and delay or prevent progressive osteoarthritis.[6]

Chondroplasty is a non-specific term that encompasses several techniques for the treatment of cartilage defects.[7] It includes

debridement and abrasion using mechanical 'shavers' and, more recently, thermal or radiofrequency techniques have also emerged despite some concerns these techniques might risk inducing localised chondrocyte death.[7–11] Recent national guidance was cautiously supportive of radiofrequency chondroplasty for the treatment of 'discrete chondral defects' based on a small number of clinical trials comparing the outcomes of mechanical and radiofrequency techniques.[12] It is not known which patients are most likely to benefit from chondroplasty procedures and when the procedure does not provide sustained benefit, knee arthroplasty is often indicated. The success rate of chondroplasty is, however, poorly understood and the proportion of patients undergoing subsequent knee arthroplasty after this intervention has been unknown.

The purpose of this study was to determine the proportion of patients undergoing knee chondroplasty procedures that subsequently receive a knee arthroplasty in the same knee, with specific focus on the proportion of patients undergoing early arthroplasty with 1 year or 2 years of chondroplasty. Factors associated with the risk of subsequent arthroplasty are reported and the relative risk in comparison to the general population determined.

## METHODS
### Data source
National Hospital Episode Statistics (HES) data was obtained (application DARS-NIC-68703) in a de-identified (pseudoanonymised) format from National Health Service (NHS) Digital.[13] HES contains a record of de-identified patient attendances at NHS hospitals in England.[13] The data is submitted by hospitals to claim payment for the services they provide and is also intended for secondary use, including research. HES includes episodes of care delivered in treatment centres (including those in the independent sector) funded by the NHS, episodes of care in England where patients are resident outside of England and privately funded patients treated within NHS England hospitals. The information recorded in the HES database includes patient demographic and residence data, primary and secondary diagnoses including comorbidities, and all procedures undertaken.

### Procedures
All HES records between 1 April 2007 and 31 March 2017 were extracted for patients undergoing arthroscopic chondroplasty. Patients undergoing previous arthroscopic knee surgery or simultaneous cruciate ligament reconstruction or microfracture in the same knee were excluded. Procedures were identified using the Classification of Surgical Operations and Procedures (OPCS-4) codes recorded within the HES data (see online supplementary appendix 1 for OPCS-4 code list).[14] All knee arthroplasty (partial or total) procedures were also identified (online supplementary appendix 2) for the whole population to enable the relative risk of knee arthroplasty with and without a history of chondroplasty to be determined.

### Outcomes
The primary outcome was knee arthroplasty, matched to the side of any previous chondroplasty (using recorded OPCS-4 laterality codes).

### Statistical analysis
Stata V.15.1 (StataCorp, College Station, Texas, USA) was used to perform all analysis. In accordance with Office for National Statistics (ONS) and NHS Digital guidance, rates where the number of events was less than six were suppressed.[15] Procedures with date errors or missing laterality were excluded. The absolute rate of knee arthroplasty was determined at 1 year, 2 years, 5 years and 8 years following arthroscopic chondroplasty as the proportion of the cohort with this minimum period of follow-up. Mortality adjusted Kaplan-Meier survival analysis (survival was defined as not undergoing knee arthroplasty) was also performed and stratified by patient age group and sex.

A Cox proportional hazards model was used first to calculate the unadjusted HR of knee arthroplasty over time by age group, sex, index of multiple deprivation (quintile derived from regional factors in England including average income, employment, education, housing and crime; 1=least deprived area, 5=most deprived), ethnicity, modified Charlson comorbidity index (derived with maximum 5-year diagnosis code lookback period),[16–18] year of treatment (chondroplasty), rurality and ethnicity, respectively.[16–19] The HRs were then adjusted including all these variables in the model.

The relative risk (risk ratio) of knee arthroplasty in the population of patients with a history of chondroplasty in comparison to the general population (without a history of chondroplasty) was estimated for the year 2016 to 2017. All patients undergoing knee arthroplasty in 2016 to 2017 were identified and the number of these patients with a recorded previous chondroplasty (in the prior 10 years of HES data), versus those without, made up the numerator for each respective population. The chondroplasty population denominator was the number of patients with a history of chondroplasty that had not undergone a knee arthroplasty prior to 2016 to 2017. The denominator for the non-chondroplasty population was the ONS mid-year population estimate less the chondroplasty population.

### Patient and public involvement
There was no patient and public involvement in this study.

## RESULTS
Over the study period, 157 730 chondroplasty patients were identified as eligible for analysis (figure 1). The mean age of the chondroplasty cohort was 51.7 year (SD 13.8) and 48.1% were female (table 1). Over the same period, 604 056 patients underwent knee arthroplasty, of

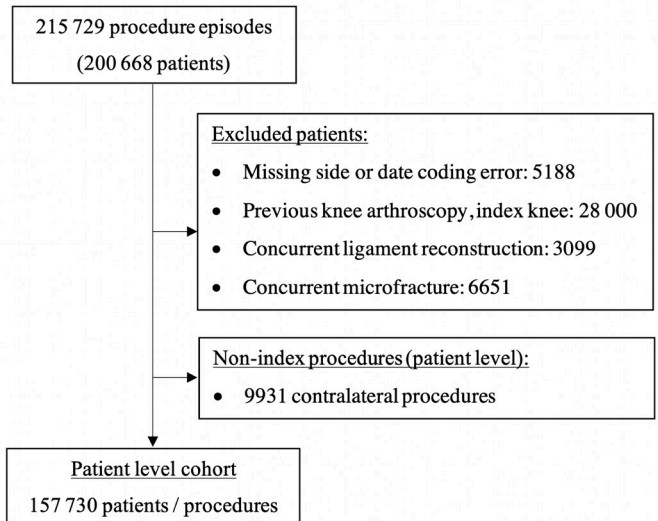

215 729 procedure episodes
(200 668 patients)

Excluded patients:
- Missing side or date coding error: 5188
- Previous knee arthroscopy, index knee: 28 000
- Concurrent ligament reconstruction: 3099
- Concurrent microfracture: 6651

Non-index procedures (patient level):
- 9931 contralateral procedures

Patient level cohort
157 730 patients / procedures

**Figure 1** Flow chart illustrating extraction of patient level cohort.

which 35 916 (5.95%) had a record of a previous chondroplasty (table 1).

Overall, following chondroplasty, 5.91% (7984/135 197; 95% CI 5.78 to 6.03) patients underwent knee arthroplasty within 1 year, 9.41% (10 787/114 592; 95% CI 9.24 to 9.58) within 2 years, 14.22% (8145/57 267; 95% CI 13.94 to 14.51) within 5 years and 17.61% (2879/16 347; 95% CI 17.03 to 18.20) within 8 years (table 2). The risk of arthroplasty was greater in female patients (adjusted HR 1.38; 95% CI 1.34 to 1.42) and in older patients (adjusted HR 1.33 per 5 years; 95% CI 1.32 to 1.34) (table 3, figure 2; figure 3). Patients with a greater comorbidity index were also at increased risk of subsequently undergoing arthroplasty (adjusted HR 1.03 per five units Charlson index; 95% CI 1.01 to 1.05).

The risk of knee arthroplasty after chondroplasty fell slightly over time, by year of chondroplasty treatment (adjusted HR 0.95 per 5 years; 95% CI 0.92 to 0.98). Patients in regions of increased deprivation and patients of white ethnicity were at greater risk of subsequent arthroplasty (table 2). Patients undergoing concurrent meniscal surgery were also at greater risk of subsequent arthroplasty (adjusted HR 1.09; 95% CI 1.06 to 1.13).

In 2016 to 2017, the rate of knee arthroplasty was 3.49% (95% CI 3.39 to 3.60) in patients (aged 30 or older) with a recorded history of chondroplasty and 0.20% (95% CI 0.19 to 0.20) in patients without a record of chondroplasty. This corresponded to an overall relative risk of knee arthroplasty for the chondroplasty cohort patients of 17.32 times (risk ratio (RR); 95% CI 16.81 to 17.84) that of the general population (table 4).

Although the absolute annual rate of knee arthroplasty was low, the relative risk of undergoing knee arthroplasty at a younger age was greatly elevated in comparison to arthroplasty at an older age, as shown in table 4. Patients aged 30 to 39 with a history of a previous chondroplasty were 170.92 times (RR; 95% CI 116.72 to 250.30) more likely to undergo knee arthroplasty than the general

population, per year, in comparison to 11.09 times (RR; 95% CI 10.42 to 11.80) more likely for the over 69 age group.

## DISCUSSION

### Principal findings

Patients undergoing chondroplasty procedures of the knee have a 17 times increased risk of receiving a knee arthroplasty compared with the general population. Nearly 10% of patients will have received a knee arthroplasty within 2 years of the chondroplasty procedure. The relative risk of undergoing arthroplasty at a young age is particularly elevated, reaching 171 times the general population rate for arthroplasty between the ages of 30 and 39 years of age. For a proportion of patients, the results indicate insufficient benefit to prevent the need for knee arthroplasty within 1 or 2 years, but the risk had these patients not undergone chondroplasty remains unknown.

### Comparison to other studies

We previously reported trends in chondroplasty surgery in England, but data from other countries is not available.[20] The age-sex standardised rate of chondroplasty increased 191% from 17.6/100 000 (95% CI 17.2 to 18.0) in 2007/2008 to 51.2/100 000 (95% CI 50.6 to 51.7) in 2016/2017.[20] The rate of chondroplasty was greatest in patients aged 40 to 59 years (increasing 210% from 34.3/100 000 in 2007/2008 to 106.4/100 000 in 2016/2017.[20]

In England, although national guidance has been cautiously supportive of radiofrequency chondroplasty for specific indications, there is only limited evidence demonstrating the effectiveness of chondroplasty compared with alternative surgical or non-surgical treatments.[12] The only randomised studies have been limited to comparisons of different chondroplasty techniques.[10 12] Long-term outcomes following chondroplasty have yet to be reported.[12]

Older patients are much more likely to have generalised osteoarthritis, rather than 'discrete chondral defects' for which the national guidance supports radiofrequency chondroplasty.[12 21] For more generalised osteoarthritis, chondroplasty is analogous to debridement and washout, where multiple clinical trials demonstrate no benefit.[2–4] The use of chondroplasty in the treatment of patients with more generalised chondral pathology is therefore unproven and not recommended.[22] In our study, there was considerable age-group variation in outcomes, with 18.8% of patients aged 60 to 79 years undergoing arthroplasty within 2 years of chondroplasty, in comparison to 0.43% for patients undergoing chondroplasty aged 20 to 39 years. This observation is consistent with the presence of more established osteoarthritis in older age groups.

Female patients were observed to be of greater risk of subsequent arthroplasty in our study. This has previously been observed following knee arthroscopy in the USA.[23]

**Table 1**  Demographics and descriptive statistics of cohort

| | Chondroplasty cohort | | Knee arthroplasty cohort | | | |
| | All cases | | No previous chondroplasty | | Previous chondroplasty | |
| | n | % | n | % | n | % |
|---|---|---|---|---|---|---|
| Total | 157 730 | 100.00 | 568 140 | 94.05 | 35 916 | 5.95 |
| Sex | | | | | | |
| Male | 81 884 | 51.91 | 244 684 | 43.07 | 15 512 | 43.19 |
| Female | 75 846 | 48.09 | 323 456 | 56.93 | 20 404 | 56.81 |
| Age group (years) | | | | | | |
| <20 | 2868 | 1.82 | 1179 | 0.21 | 1 | <0.01 |
| 20–39 | 24 648 | 15.63 | 1568 | 0.28 | 353 | 0.98 |
| 40–59 | 83 258 | 52.79 | 85 797 | 15.1 | 14 023 | 39.04 |
| 60–79 | 45 191 | 28.65 | 400 541 | 70.5 | 20 361 | 56.69 |
| 80+ | 1765 | 1.12 | 79 055 | 13.91 | 1178 | 3.28 |
| Charlson comorbidity index | | | | | | |
| 0 | 121 605 | 77.10 | 534 399 | 94.06 | 27 331 | 76.1 |
| 1–15 | 34 719 | 22.01 | 31 683 | 5.58 | 8175 | 22.76 |
| 16–30 | 1296 | 0.82 | 1879 | 0.33 | 393 | 1.09 |
| 31–50 | 110 | 0.07 | 179 | 0.03 | 17 | 0.05 |
| Index of multiple deprivation (quintiles) | | | | | | |
| 1=least deprived | 36 043 | 23.21 | 121 813 | 21.44 | 7921 | 22.05 |
| 2 | 35 189 | 22.66 | 127 672 | 22.47 | 7938 | 22.1 |
| 3 | 32 493 | 20.92 | 123 160 | 21.68 | 7806 | 21.73 |
| 4 | 27 312 | 17.59 | 103 236 | 18.17 | 6372 | 17.74 |
| 5=most deprived | 24 266 | 15.62 | 85 283 | 15.01 | 5416 | 15.08 |
| Missing | 2427 | | 6976 | | 463 | |
| Rurality | | | | | | |
| Urban | 119 766 | 76.42 | 423 895 | 74.61 | 27 157 | 75.61 |
| Rural | 36 953 | 23.58 | 141 271 | 24.87 | 8634 | 24.04 |
| Missing | 1011 | | 2974 | | 125 | |
| Ethnicity | | | | | | |
| White | 141 928 | 94.43 | 525 934 | 92.57 | 34 349 | 95.64 |
| Mixed | 953 | 0.63 | 1844 | 0.32 | 115 | 0.32 |
| Asian | 4511 | 3.00 | 19 203 | 3.38 | 804 | 2.24 |
| Black | 2122 | 1.41 | 5840 | 1.03 | 193 | 0.54 |
| Other | 792 | 0.53 | 1367 | 0.24 | 68 | 0.19 |
| Missing | 7424 | | 13 952 | | 387 | |
| Concurrent procedures | | | | | | |
| None | 65 987 | 41.84 | – | – | – | – |
| Meniscal | 91 743 | 58.16 | – | – | – | – |

Patients of white ethnicity and greater deprivation were also at greater risk in our cohort. These findings may reflect differences in healthcare access including treatment thresholds for either the chondroplasty or knee arthroplasty, or differences in care seeking behaviour which has been shown to be influenced by socioeconomic, cultural, occupational and psychological factors, or there could be biological factors underlying the observation.[24–26]

Patients with a greater comorbidity index were more likely to undergo subsequent arthroplasty, and the reason for this is unclear. One possible explanation might be an association between comorbidity and higher body mass index (BMI), which is not recorded in this data set, with patients having a greater BMI being more likely to progress to end-stage osteoarthritis, or that these patients had more severe pathology at the time of their index chondroplasty.[27]

**Table 2** Cohort demographics and adjusted odds of arthroplasty

| | 1-year outcome* | | | 2-year outcome* | | | 5-year outcome* | | | 8-year outcome* | | |
|---|---|---|---|---|---|---|---|---|---|---|---|---|
| | n | n TKA | % (95% CI) | n | n TKA | % (95% CI) | n | n TKA | % (95% CI) | n | n TKA | % (95% CI) |
| Total | 135 197 | 7984 | 5.91% (5.78 to 6.03) | 114 592 | 10 787 | 9.41% (9.24 to 9.58) | 57 267 | 8145 | 14.22% (13.94 to 14.51) | 16 347 | 2879 | 17.61% (17.03 to 18.20) |
| **Sex** | | | | | | | | | | | | |
| Male | 69 787 | 3160 | 4.53% (4.37 to 4.68) | 59 101 | 4261 | 7.21% (7.00 to 7.42) | 29 688 | 3315 | 11.17% (10.81 to 11.53) | 8514 | 1208 | 14.19% (13.45 to 14.95) |
| Female | 65 410 | 4824 | 7.38% (7.18 to 7.58) | 55 491 | 6526 | 11.76% (11.49 to 12.03) | 27 579 | 4830 | 17.51% (17.07 to 17.97) | 7833 | 1671 | 21.33% (20.43 to 22.26) |
| **Age group (years)** | | | | | | | | | | | | |
| <20 | – | – | – | – | – | – | – | – | – | – | – | – |
| 20–39 | 21 548 | 48 | 0.22% (0.16 to 0.30) | 18 583 | 79 | 0.43% (0.34 to 0.53) | 10 004 | 95 | 0.95% (0.77 to 1.16) | 3094 | 53 | 1.71% (1.29 to 2.23) |
| 40–59 | 72 345 | 2654 | 3.67% (3.53 to 3.81) | 60 974 | 4049 | 6.64% (6.44 to 6.84) | 29 844 | 3327 | 11.15% (10.79 to 11.51) | 8552 | 1287 | 15.05% (14.30 to 15.82) |
| 60–79 | 39 741 | 4994 | 12.57% (12.24 to 12.90) | 33 680 | 6343 | 18.83% (18.42 to 19.25) | 16 716 | 4522 | 27.05% (26.38 to 27.73) | 4508 | 1479 | 32.81% (31.44 to 34.20) |
| 80 + | 1563 | 288 | 18.43% (16.53 to 20.44) | 1355 | 316 | 23.32% (21.09 to 25.67) | 703 | 201 | 28.59% (25.28 to 32.09) | 193 | 60 | 31.09% (24.64 to 38.13) |
| **Charlson comorbidity index** | | | | | | | | | | | | |
| 0 | 104 530 | 5369 | 5.14% (5.00 to 5.27) | 89 081 | 7366 | 8.27% (8.09 to 8.45) | 45 505 | 5837 | 12.83% (12.52 to 13.14) | 13 362 | 2148 | 16.08% (15.46 to 16.71) |
| 1–15 | 29 475 | 2467 | 8.37% (8.06 to 8.69) | 24 540 | 3228 | 13.15% (12.73 to 13.58) | 11 371 | 2207 | 19.41% (18.69 to 20.15) | 2884 | 701 | 24.31% (22.75 to 25.91) |
| 16–30 | 1102 | 138 | 12.52% (10.63 to 14.62) | 898 | 180 | 20.04% (17.47 to 22.82) | 391 | 101 | 25.83% (21.56 to 30.47) | 101 | 30 | 29.70% (21.02 to 39.61) |
| 31–50 | 90 | 10 | 11.11% (5.46 to 19.49) | 73 | 13 | 17.81% (9.84 to 28.53) | – | – | – | – | – | – |
| **Index of multiple deprivation (quintiles)** | | | | | | | | | | | | |
| 1 | 31 054 | 1846 | 5.94% (5.68 to 6.21) | 26 546 | 2405 | 9.06% (8.72 to 9.41) | 13 422 | 1835 | 13.67% (13.09 to 14.26) | 3878 | 629 | 16.22% (15.07 to 17.42) |
| 2 | 30 218 | 1799 | 5.95% (5.69 to 6.23) | 25 638 | 2409 | 9.40% (9.04 to 9.76) | 12 819 | 1779 | 13.88% (13.28 to 14.49) | 3474 | 617 | 17.76% (16.50 to 19.07) |
| 3 | 27 974 | 1737 | 6.21% (5.93 to 6.50) | 23 721 | 2324 | 9.80% (9.42 to 10.18) | 11 833 | 1772 | 14.98% (14.34 to 15.63) | 3374 | 635 | 18.82% (17.51 to 20.18) |
| 4 | 23 312 | 1387 | 5.95% (5.65 to 6.26) | 19 702 | 1913 | 9.71% (9.30 to 10.13) | 9771 | 1420 | 14.53% (13.84 to 15.25) | 2762 | 496 | 17.96% (16.54 to 19.44) |

Continued

**Table 2** Continued

| | 1-year outcome* | | | 2-year outcome* | | | 5-year outcome* | | | 8-year outcome* | | |
|---|---|---|---|---|---|---|---|---|---|---|---|---|
| | n | n TKA | % (95% CI) | n | n TKA | % (95% CI) | n | n TKA | % (95% CI) | n | n TKA | % (95% CI) |
| 5 | 20 591 | 1104 | 5.36% (5.06 to 5.68) | 17 194 | 1588 | 9.24% (8.81 to 9.68) | 8420 | 1197 | 14.22% (13.48 to 14.98) | 2451 | 444 | 18.12% (16.61 to 19.70) |
| **Rurality** | | | | | | | | | | | | |
| Urban | 102 665 | 6004 | 5.85% (5.71 to 5.99) | 86 807 | 8135 | 9.37% (9.18 to 9.57) | 43 287 | 6148 | 14.20% (13.88 to 14.54) | 12 242 | 2154 | 17.60% (16.92 to 18.28) |
| Rural | 31 760 | 1944 | 6.12% (5.86 to 6.39) | 27 127 | 2613 | 9.63% (9.28 to 9.99) | 13 739 | 1980 | 14.41% (13.83 to 15.01) | 3987 | 713 | 17.88% (16.70 to 19.11) |
| **Ethnicity** | | | | | | | | | | | | |
| White | 122 261 | 7672 | 6.28% (6.14 to 6.41) | 103 979 | 10 366 | 9.97% (9.79 to 10.15) | 52 267 | 7834 | 14.99% (14.68 to 15.30) | 14 908 | 2750 | 18.45% (17.83 to 19.08) |
| Mixed | 750 | 21 | 2.80% (1.74 to 4.25) | 609 | 32 | 5.25% (3.62 to 7.34) | 278 | 22 | 7.91% (5.03 to 11.74) | 76 | 8 | 10.53% (4.66 to 19.69) |
| Asian | 3722 | 130 | 3.49% (2.93 to 4.13) | 3088 | 186 | 6.02% (5.21 to 6.92) | 1465 | 167 | 11.40% (9.82 to 13.14) | 362 | 75 | 20.72% (16.66 to 25.26) |
| Black | 1770 | 27 | 1.53% (1.01 to 2.21) | 1466 | 53 | 3.62% (2.72 to 4.70) | 677 | 38 | 5.61% (4.00 to 7.62) | 171 | 15 | 8.77% (4.99 to 14.06) |
| Other | 645 | 15 | 2.33% (1.31 to 3.81) | 518 | 18 | 3.47% (2.07 to 5.44) | 250 | 21 | 8.40% (5.27 to 12.55) | 75 | 9 | 12.00% (5.64 to 21.56) |
| **Concurrent procedures** | | | | | | | | | | | | |
| None | 57 208 | 2686 | 4.70% (4.52 to 4.87) | 50 256 | 3754 | 7.47% (7.24 to 7.70) | 28 578 | 3252 | 11.38% (11.01 to 11.75) | 9370 | 1389 | 14.82% (14.11 to 15.56) |
| Meniscal | 77 989 | 5298 | 6.79% (6.62 to 6.97) | 64 336 | 7033 | 10.93% (10.69 to 11.18) | 28 689 | 4893 | 17.06% (16.62 to 17.50) | 6977 | 1490 | 21.36% (20.40 to 22.34) |

- =suppressed due to small numbers
*Excluding those patients where the date of their procedure was less than this number of years from the end of the observation period in the data set.
TKA, total or partial knee arthroplasty.

**Table 3** Unadjusted and adjusted* risk of knee arthroplasty following arthroscopic chondroplasty

| | Unadjusted risk subsequent TKA | | Adjusted risk subsequent TKA | |
|---|---|---|---|---|
| | HR | 95% CI | HR | 95% CI |
| **Sex** | | | | |
| Male | 1.00 | 1.00 | 1.00 | 1.00 |
| Female | 1.61 | 1.57 to 1.66 | 1.38 | 1.34 to 1.42 |
| **Age (per 5 years)** | | | | |
| Per year | 1.35 | 1.35 to 1.36 | 1.33 | 1.32 to 1.34 |
| **Year of treatment (per 5 years)** | | | | |
| Year | 0.99 | 0.96 to 1.03 | 0.95 | 0.92 to 0.98 |
| **Charlson comorbidity index (per five units)** | | | | |
| Charlson index | 1.29 | 1.27 to 1.31 | 1.03 | 1.01 to 1.05 |
| **Index of multiple deprivation (quintile)** | | | | |
| 1=least | 1.00 | 1.00 | 1.00 | 1.00 |
| 2 | 1.03 | 0.99 to 1.08 | 1.07 | 1.03 to 1.12 |
| 3 | 1.08 | 1.04 to 1.13 | 1.17 | 1.12 to 1.22 |
| 4 | 1.03 | 0.99 to 1.08 | 1.20 | 1.15 to 1.26 |
| 5=most | 1.01 | 0.96 to 1.06 | 1.29 | 1.23 to 1.36 |
| **Rurality** | | | | |
| Urban | 1.00 | 1.00 | 1.00 | 1.00 |
| Rural | 1.03 | 1.00 to 1.07 | 0.99 | 0.95 to 1.02 |
| **Ethnicity** | | | | |
| White | 1.00 | 1.00 | 1.00 | 1.00 |
| Mixed | 0.50 | 0.38 to 0.65 | 0.66 | 0.51 to 0.86 |
| Asian | 0.65 | 0.59 to 0.72 | 0.73 | 0.66 to 0.81 |
| Black | 0.35 | 0.28 to 0.42 | 0.44 | 0.36 to 0.54 |
| Other | 0.34 | 0.24 to 0.48 | 0.45 | 0.32 to 0.64 |
| **Concurrent procedures** | | | | |
| None | 1.00 | 1.00 | 1.00 | 1.00 |
| Meniscal surgery | 1.52 | 1.48 to 1.57 | 1.09 | 1.06 to 1.13 |

*Adjusted by all variables in the table.
†Age <20 years suppressed due to small numbers.
TKA, total or partial knee arthroplasty.

Patients undergoing concurrent meniscal surgery were also more likely to undergo subsequent arthroplasty, which is expected given the association between meniscal injury, osteoarthritis and knee arthroplasty.[28]

Recently, there has been renewed focus on the importance of and requirements for individualised patient consent.[29] Our findings make an important contribution to the current evidence, and patients can now be appropriately counselled and consented with knowledge of anticipated long-term outcomes.

### Strength and limitations

A key strength of our study is the identification of all knee chondroplasty procedures performed in the National

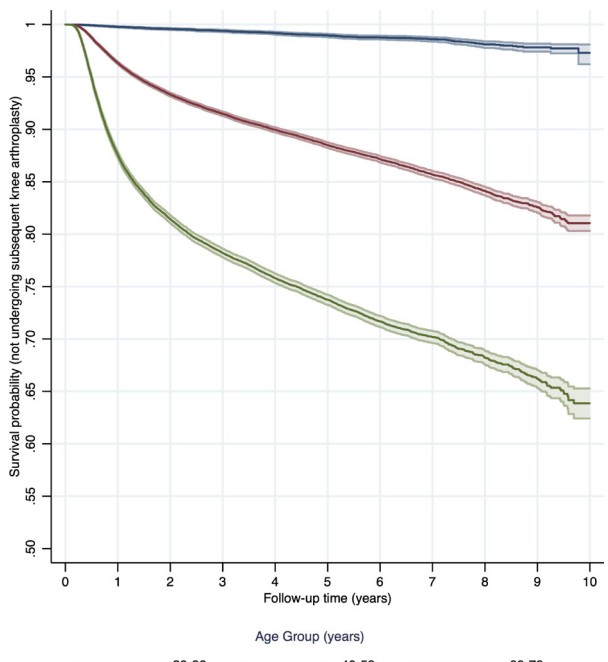

**Figure 2** Survival curve (not undergoing knee arthroplasty) following chondroplasty by age[+]
Age groups < 20 years and 80+ years suppressed due to small numbers

Health Service over a 10-year period, creating the largest reported cohort of patients receiving this procedure. Patients with a history of prior arthroscopy in the same

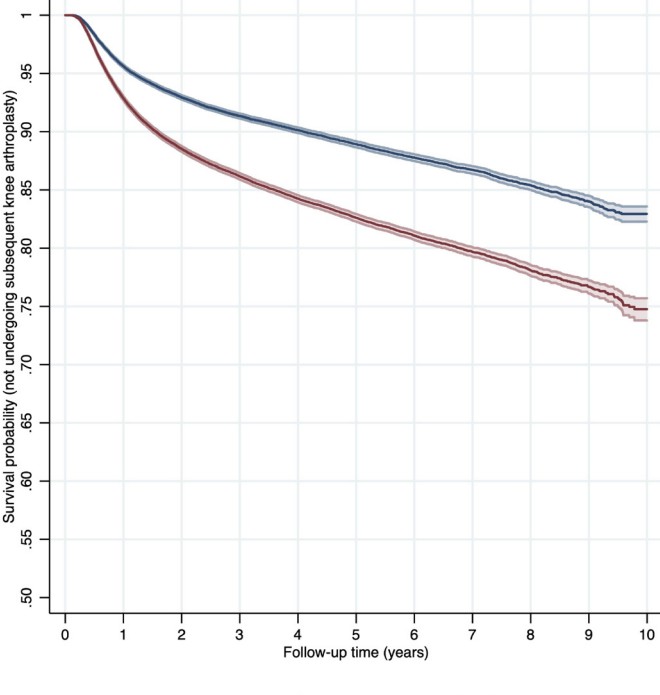

**Figure 3** Survival curve (not undergoing knee arthroplasty) following chondroplasty by sex.
Age groups < 20 years and 80+ years suppressed due to small numbers

**Table 4** Rates and relative risk of undergoing TKA with previous chondroplasty by age at TKA in 2016 to 2017

| Age at TKA (years) | Prior chondroplasty | | Without prior chondroplasty | | Relative risk | |
|---|---|---|---|---|---|---|
| | Annual rate TKA/100 k | 95% CI | Annual rate TKA/100 k | 95% CI | RR | 95% CI |
| 30–39 | 274.48 (0.27%) | 190.16 to 383.35 (0.19% to 0.38%) | 1.60 (0.00%) | 1.32 to 1.92 (0.00% to 0.00%) | 170.92 | 116.72 to 250.30 |
| 40–49 | 1454.02 (1.45%) | 1318.04 to 1600.06 (1.32% to 1.60%) | 19.79 (0.02%) | 18.79 to 20.82 (0.02% to 0.02%) | 72.45 | 65.00 to 80.76 |
| 50–59 | 3626.62 (3.63%) | 3448.20 to 3811.60 (3.45% to 3.81%) | 130.68 (0.13%) | 128.05 to 133.35 (0.13% to 0.13%) | 26.82 | 25.41 to 28.30 |
| 60–69 | 5179.17 (5.18%) | 4933.67 to 5433.20 (4.93% to 5.43%) | 386.68 (0.39%) | 381.64 to 391.77 (0.38% to 0.39%) | 12.78 | 12.16 to 13.44 |
| 70+ | 6090.50 (6.09%) | 5721.53 to 6475.82 (5.72% to 6.48%) | 520.46 (0.52%) | 514.89 to 526.07 (0.51% to 0.53%) | 11.09 | 10.42 to 11.80 |
| Overall (30+) | 3494.61 (3.49%) | 3394.82 to 3596.52 (3.39% to 3.60%) | 195.38 (0.20%) | 193.90 to 196.87 (0.19% to 0.20%) | 17.32 | 16.81 to 17.84 |

TKA, total or partial knee arthroplasty; RR, risk ratio.

knee, simultaneous ligament reconstruction or microfracture were excluded as potential confounding factors. It should still be noted that patients undergoing non-NHS treatment, for example, knee arthroplasty in the private sector after a previous knee arthroscopy under NHS care, would not be captured in this data set and the number of these procedures performed in the private sector is currently unknown. National data does indicate, however, that private healthcare expenditure as a proportion of total healthcare expenditure has remained relatively stable at around 17% to 18% of total health expenditure between 2005 and 2015.[30] For all observational studies utilising large data sets there may be some concerns raised about coding accuracy. The data in our study was cleaned prior to analysis, excluding patients where procedures were missing the side of intervention and cases where date coding errors were identified. Although some other data coding errors are inevitable, data errors in procedure coding would result in hospitals not receiving payment for surgery performed, and this provides a strong incentive for data accuracy with regards to the coding data analysed in this study.

We were able to stratify risk of arthroplasty by a large number of patient factors, but certain procedure specific data is not recorded. Operative factors, such as the affected compartment of the knee and extent of initial cartilage damage before intervention, are not recorded in this database. These factors may be important in determining outcome, for example, there are likely to be differences in long-term outcomes between chondroplasty performed to the tibiofemoral joint in comparison to the patellofemoral joint.[31] Other unmeasured sources of potential confounding include BMI, leg alignment and radiographic status at the time of intervention. These are important considerations when considering if a patient is suitable for chondral surgery intervention, but the specific impact of these factors on long-term

outcomes in this population remains uncertain. Subjective, patient-reported, symptomatic outcome data is not yet available for this cohort and radiographic outcomes are not recorded in the HES database. Instead, our study focussed on the objective, measurable outcome of knee arthroplasty, matched to the same knee as the previous chondroplasty surgery intervention. Although knee arthroplasty represents the end-stage of symptomatic failure for patients with osteoarthritis, it is likely to considerably underestimate the overall health and symptom burden in this cohort. Patients, particularly younger patients, may not have been willing or suitable candidates for knee arthroplasty, and the threshold for arthroplasty may have been much higher for younger age groups or older patients with multiple comorbidities. It is also important to note that, in general, 'chondroplasty' is a non-specific term that encompasses several techniques for the debridement of cartilage defects.[7] The findings in this paper cannot be generalised to other types of arthroscopic and joint preservation surgery, cartilage repair and regeneration techniques, such as microfracture and autologous chondrocyte implantation.[6 31]

Our study represents a high-risk cohort of patients with cartilage damage. It is unknown from this observational data whether undergoing the chondroplasty procedure was beneficial to the symptoms or prognosis of these individuals over the full study period. That is, it is not known whether the chondroplasty procedure delayed or prevented arthroplasty in those patients that did not undergo arthroplasty (approximately 86% by 5 years), in which case delivery of the intervention may have been cost-effective, or the converse interpretation is that the procedure may have been overused and that the natural history of symptomatic osteoarthritis in this population was unaltered. For example, the observed proportion of patients undergoing arthroplasty within 1 year of their arthroscopic chondroplasty (6%) is suggestive of

suboptimal treatment selection. These individuals are highly unlikely to have had only localised or partial thickness lesions and our results may indicate that knee arthroplasty may have been a more appropriate treatment. Nevertheless, the symptomatic outcome in the patients that did not undergo arthroplasty is not known and the answer to whether the procedure is cost-effective with optimal patient selection is unknown and requires evaluation in a high-quality randomised controlled trial with a non-operative treatment arm. Such a trial should help to evaluate the optimal indications for chondroplasty, assess the relative rate of progression of treated chondral damage with versus without chondroplasty and ultimately determine whether appropriate use of chondroplasty is beneficial to patient outcome including, potentially, the long-term demand for knee arthroplasty.

Our study reports the long-term outcomes following chondroplasty in a high-risk cohort of patients with cartilage damage for the first time. Our findings stratified by a range of patient-specific factors however further work is required to optimise treatment selection and additional patient information may allow more accurate prediction of outcome and guide clinical management.

## CONCLUSION

The risk of knee arthroplasty is 17 times greater in patients with a history of knee chondroplasty and in a proportion of patients, there is insufficient benefit to prevent the need for knee arthroplasty within 1 or 2 years. These important new data help inform patients and clinicians of the long-term outcomes following this procedure, at the population level, for the first time. Enhanced clinical guidance on the appropriate indications for chondroplasty are required and there is a need for high-quality randomised studies to determine the relative clinical and cost-effectiveness of this intervention in comparison to alternative, including non-surgical, treatments.

**Contributors** SGFA: concept, methodology, analysis, writing and editing paper, guarantor. AJRP: writing and editing paper. AJ: methodology, analysis and editing paper. DB: concept and editing paper. AJP: concept, methodology and editing paper.

**Funding** The authors have not declared a specific grant for this research from any funding agency in the public, commercial or not-for-profit sectors.

**Competing interests** AJ has received consultancy fees from Freshfields Bruckhaus Deringer (on behalf of Smith & Nephew Orthopaedics Limited), and is a member of the Data Safety and Monitoring Board (which involved receipt of fees) from Anthera Pharmaceuticals, Inc. All other authors declare no financial relationships with any organisations that might have an interest in the submitted work in the previous 3 years, no other relationships or activities that could appear to have influenced the submitted work.

**Patient consent for publication** Not required.

**Provenance and peer review** Not commissioned; externally peer reviewed.

**Data availability statement** No additional data is available.

**ORCID iDs**
Simon G F Abram http://orcid.org/0000-0002-4452-6499
Andrew J Price http://orcid.org/0000-0002-4258-5866

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
