## [Reviewer comments · BMJ Open]

ARTICLE DETAILS

TITLE (PROVISIONAL)	Rates of knee arthroplasty in patients with a history of arthroscopic chondroplasty: results from a retrospective cohort study utilising the National Hospital Episode Statistics for England
AUTHORS	Abram, Simon; Palmer, Antony; Judge, Andrew; Beard, David; Price, Andrew

VERSION 1 – REVIEW

REVIEWER	Feng-Chih Kuo Kaohsiung Chang Gung Memorial Hospital, Kaohsiung, Taiwan
REVIEW RETURNED	14-Apr-2019

GENERAL COMMENTS	Thank you for having me an opportunity to review the manuscript entitled: " Rates of knee arthroplasty in patients with a history of arthroscopic chondroplasty: results from a retrospective cohort study utilising the National Hospital Episode Statistics for England (bmjopen-2019-030609)," which has been submitted to The BMJ Open. The authors used the National Hospital Episode Statistics (HES) database to evaluate the effect of arthroscopic chondroplasty in patients with chondral defects. They used patients who had knee arthroplasty eventually as their primary outcome and did a mortality-adjusted survival analysis. They found patients with a history of arthroscopic chondroplasty had a higher RR of undergoing TKA than those without prior chondroplasty in the normal population. Patients ≥ 30 years with a history of chondroplasty had a higher RR of 17.32 undergoing TKA than the general population without a history of chondroplasty. This is a well-written and comprehensive study, but I found some study flaw and have comments about this study. 1. The major study flaw was the comparison group. The authors used patients without a prior chondroplasty in normal population as a control group. However, the readers could not distinguish the effect of arthroscopic chondroplasty in the treatment of chondral defects compared with those without arthroscopic chondroplasty treatment. But the author used a normal population without a prior chondroplasty as their control group. The risk of subsequent TKA would be higher in the study group because of different unequal patients' population.2. Why did you use the TKA cohort in 2016-2017? What if patients underwent TKA before 2015? For my viewpoint, I think they should use the same study period or longer period to determine which patients underwent TKA, not only in 2016-2017.3. In the introduction section, they should have a better transition sentence from the current study to their questions. For example, what was the current success rate for arthroscopic chondroplasty?
---

	What was the rate of subsequent TKA following arthroscopic chondroplasty? Was there different for the rate of subsequent TKA in patients with a chondral defect with or without the treatment of arthroscopic chondroplasty? 4. In the result section, the number (157,830 chondroplasty patients) did not match the number (157,730) in table 1. Were any bilateral chondroplasty procedures? And in table 2, the number of receiving chondroplasty was 135,197. Where were the lost 22,533 patients? Did they die within one year following arthroscopic surgery? 5. Figure 2 and figure 3 did not mark the meaning of each survival curve. 6. In the result section, line 40-41 and line 49, the description should be table 4, not table 3.
--	---

REVIEWER	William B Weeks Microsoft, US
REVIEW RETURNED	25-Sep-2019

GENERAL COMMENTS	This is an interesting study that examines the relative risk of receiving same-side TKA following chondroplasty. It finds that the risk is high, compared to that of the general population (without a history of chondroplasty), with the relative risk being highest for younger patients and declining with age. Risk increases over time since chondroplasty, with TKA within 1 year of chondroplasty being 6% and that within 8 years being about triple that. The study is driven by a concern that chondroplasty is an ineffective treatment. The authors recognize that an RCT with a non-treatment arm is the only way to get at this question. While TKA following chondroplasty can be seen as chondroplasty failure, chondroplasty advocates might use the authors' study to retort that, at 8 years out, over 80% of chondroplasty patients were able to avoid (a more costly, intrusive, and risky) TKA procedure. Given that volumes of chondroplasty and TKA seemingly go in opposite directions with respect to age (falling for the former and rising for the latter), it does seem that chondroplasty might be considered a way to try something less intrusive, particularly among those less likely to have osteoarthritis, to address current needs and, hopefully, stave off a more invasive procedure. And, of course, that might not work; but after 8 years only 1.7% of the youngest age group got a TKA. The authors, it seems, anticipates this retort and compares rates of TKA among chondroplasty recipients to TKA rates among non-chondroplasty recipients. The key, of course, is the comparison group; for instance, in Table 1, we do not get demographics for the comparison group (those with TKA but without chondroplasty). And, of course, those without prior chondroplasty might have a dramatically lower prevalence of knee osteo-arthritis (which the authors try to address, in their discussion, noting that older patients are more likely to have osteoarthritis (and get TKA, which is why the addition to Table 1 would be helpful – are the distributions the same?)). Without knowing WHY patients got their chondroplasty, WHETHER it was warranted (and the patient did NOT have osteoarthritis), and WHETHER it was meant as a (largely successful, if warranted) attempt to forgo a more costly and invasive procedure it is fairly difficult to draw conclusions here, other than that an RCT is needed. So I am torn, regarding the manuscript.
---

	The observational nature in the context of these massively unknown aspects make the findings not very valuable: it is information, but reasonable ways to interpret the data abound. And the authors have already suggested overuse of chondroplasty in the UK, though that seemingly is declining. Nonetheless, this very lack of clarity seemingly warrants an RCT. If the paper is to be revised, I'd ask for:  1. Addition of the non-chondroplasty TKAs AND non-chondroplasty non-TKAs to Table 1. 2. An estimate of missing numbers of chondroplasty and TKA outside of the NHS. 3. A legend for the figures – what is red and blue in each, for instance? 4. Page 8 line 31 has an incomplete sentence “In England, although...”
--	---

VERSION 1 – AUTHOR RESPONSE

Reviewer: 1

Reviewer Name: Feang-Chih Kuo

Institution and Country: Kaohsiung Chang Gung Memorial Hospital, Kaohsiung, Taiwan

Please state any competing interests or state 'None declared': None

Please leave your comments for the authors below

Dear Editor

Thank you for having me an opportunity to review the manuscript entitled: " Rates of knee arthroplasty in patients with a history of arthroscopic chondroplasty: results from a retrospective cohort study utilising the National Hospital Episode Statistics for England (bmjopen-2019-030609)," which has been submitted to The BMJ Open.

The authors used the National Hospital Episode Statistics (HES) database to evaluate the effect of arthroscopic chondroplasty in patients with chondral defects. They used patients who had knee arthroplasty eventually as their primary outcome and did a mortality-adjusted survival analysis. They found patients with a history of arthroscopic chondroplasty had a higher RR of undergoing TKA than those without prior chondroplasty in the normal population. Patients ≥ 30 years with a history of chondroplasty had a higher RR of 17.32 undergoing TKA than the general population without a history of chondroplasty. This is a well-written and comprehensive study, but I found some study flaw and have comments about this study.

Thank you for this summary and comments. Responses follow to the items below.

1. The major study flaw was the comparison group. The authors used patients without a prior chondroplasty in normal population as a control group. However, the readers could not distinguish the effect of arthroscopic chondroplasty in the treatment of chondral defects compared with those without arthroscopic chondroplasty treatment. But the author used a normal population without a prior chondroplasty as their control group. The risk of subsequent TKA would be higher in the study group because of different unequal patients' population.

We agree and fully acknowledge this throughout the manuscript. This paper is descriptive and highlights associations, but causation cannot be attributed. We believe our findings will be important for patients who do undergo chondroplasty, to provide some context in consultations around the expected long-term outcomes. We have also added to Table 1 the demographics of the knee arthroplasty cohort – for both those patients with a history of a previous chondroplasty procedure and those without. We hope that this change facilitates comparison of the cohort group demographics.

2. Why did you use the TKA cohort in 2016-2017? What if patients underwent TKA before 2015? For my viewpoint, I think they should use the same study period or longer period to determine which patients underwent TKA, not only in 2016-2017.

We used the whole max 10-years study period for the survival analysis (Figure 2 etc), however for the comparative analysis of the population with vs. without a history of chondroplasty it is necessary to pick a fixed period of time. In this case, given the large dataset available, it was decided that it would be optimal to use the most recent complete year of data that was available. Patients who had undergone TKA prior to 2016-17 were identified and excluded from this comparative analysis (as the outcome occurred in a different time period from the analysis period chosen), as were the mortality cases.

3. In the introduction section, they should have a better transition sentence from the current study to their questions. For example, what was the current success rate for arthroscopic chondroplasty? What was the rate of subsequent TKA following arthroscopic chondroplasty? Was there different for the rate of subsequent TKA in patients with a chondral defect with or without the treatment of arthroscopic chondroplasty?

We have added an additional sentence to the introduction to highlight these areas of uncertainty as suggested.

4. In the result section, the number (157,830 chondroplasty patients) did not match the number (157,730) in table 1. Were any bilateral chondroplasty procedures? And in table 2, the number of receiving chondroplasty was 135,197. Where were the lost 22,533 patients? Did they die within one year following arthroscopic surgery?

Thank you – this was a typo in the results section – the correct number is 157,730 throughout as in Table 1 and Figure 1. The number 135,197 represents all those cases with at least 1-year of follow-up (in practice this means excluding patients undergoing chondroplasty after 1st April 2016 (to get a true estimate of the 1-year arthroplasty rate, at least 1-year of follow-up is required). The same applies for the 2-year, 5-year, 8-year groups in Table 2. In other words, the numbers exclude those patients where the date of their procedure was less than this number of years from the end of the observation period in the dataset. This further explanation has been added as a footnote to Table 2 to help readers to understand this.

5. Figure 2 and figure 3 did not mark the meaning of each survival curve.

Apologies it seems the electronic upload requirements resulted in these legends being lost. These are now embedded and included with this revision.

6. In the result section, line 40-41 and line 49, the description should be table 4, not table 3.

Thank you – corrected.

Reviewer: 2

Reviewer Name: William B Weeks

Institution and Country: Microsoft, US

Please state any competing interests or state 'None declared': None declared

Please leave your comments for the authors below

This is an interesting study that examines the relative risk of receiving same-side TKA following chondroplasty. It finds that the risk is high, compared to that of the general population (without a history of chondroplasty), with the relative risk being highest for younger patients and declining with age. Risk increases over time since chondroplasty, with TKA within 1 year of chondroplasty being 6% and that within 8 years being about triple that.

The study is driven by a concern that chondroplasty is an ineffective treatment. The authors

recognize that an RCT with a non-treatment arm is the only way to get at this question. While TKA following chondroplasty can be seen as chondroplasty failure, chondroplasty advocates might use the authors' study to retort that, at 8 years out, over 80% of chondroplasty patients were able to avoid (a more costly, intrusive, and risky) TKA procedure. Given that volumes of chondroplasty and TKA seemingly go in opposite directions with respect to age (falling for the former and rising for the latter), it does seem that chondroplasty might be considered a way to try something less intrusive, particularly among those less likely to have osteoarthritis, to address current needs and, hopefully, stave off a more invasive procedure. And, of course, that might not work; but after 8 years only 1.7% of the youngest age group got a TKA.

The authors, it seems, anticipates this retort and compares rates of TKA among chondroplasty recipients to TKA rates among non-chondroplasty recipients. The key, of course, is the comparison group; for instance, in Table 1, we do not get demographics for the comparison group (those with TKA but without chondroplasty). And, of course, those without prior chondroplasty might have a dramatically lower prevalence of knee osteo-arthritis (which the authors try to address, in their discussion, noting that older patients are more likely to have osteoarthritis (and get TKA, which is why the addition to Table 1 would be helpful – are the distributions the same?)).

Thank you for this careful summary and for these suggestions which we have addressed with the itemised summary below.

Without knowing WHY patients got their chondroplasty, WHETHER it was warranted (and the patient did NOT have osteoarthritis), and WHETHER it was meant as a (largely successful, if warranted) attempt to forgo a more costly and invasive procedure it is fairly difficult to draw conclusions here, other than that an RCT is needed.

So I am torn, regarding the manuscript.

The observational nature in the context of these massively unknown aspects make the findings not very valuable: it is information, but reasonable ways to interpret the data abound. And the authors have already suggested overuse of chondroplasty in the UK, though that seemingly is declining. Nonetheless, this very lack of clarity seemingly warrants an RCT.

Thank you. We have been careful in our conclusions and statements to this effect. These are descriptive findings, but we cannot determine the effectiveness of chondroplasty versus alternative interventions or no intervention for which an RCT will be necessary. Our findings are nevertheless important and will be useful to clinicians and patients in the shared-decision making and consent process – for those patients undergoing chondroplasty – and may also be helpful in the design of any future RCT.

If the paper is to be revised, I'd ask for:

1. Addition of the non-chondroplasty TKAs AND non-chondroplasty non-TKAs to Table 1.

Thank you for this suggestion. We have added the requested demographic data to Table 1.

2. An estimate of missing numbers of chondroplasty and TKA outside of the NHS.

Private health data is not available for analysis however previous studies have suggested around 17-18% of total healthcare expenditure is in the private sector. Some cases may have received chondroplasty in the NHS then arthroplasty in the private sector, but a similar proportion may have received chondroplasty in the private sector then arthroplasty in the NHS. The true numbers are unknown and this is a limitation we had already discussed but in response to your comments we have added a note about the proportion of healthcare expenditure in the private sector to the strengths and limitations section.

3. A legend for the figures – what is red and blue in each, for instance?

Apologies it seems the electronic upload requirements resulted in these legends being lost. These are now embedded and included with this revision.

4. Page 8 line 31 has an incomplete sentence “In England, although....”

Thank you for highlighting this – corrected.

VERSION 2 – REVIEW

REVIEWER	Feng-Chih Kuo Kaohsiung Chang Gung Memorial Hospital, Kaohsiung, Taiwan
REVIEW RETURNED	28-Oct-2019

GENERAL COMMENTS	The authors did a thorough response to the revised manuscripts. I have no more comments on it. I agree with it being published in BMJ Open.
---

REVIEWER	William B Weeks, MD, PhD, MBA Microsoft, USA
REVIEW RETURNED	29-Oct-2019

GENERAL COMMENTS	Thanks to the authors for addressing my concerns - primarily that demographics of the comparisons group (TKA with and without prior chondroplasty) had not been included in Table 1. Those data suggest to me that - IF CHONDROPLASTY IS WARRANTED - chondroplasty is pretty effective - only about 5% of TKAs had a prior chondroplasty, so if the narrative is that chondroplasty intervention AVOIDS later (more expensive and intrusive) TKA. A 5% failure rate over a decade isn't bad for a relatively minor procedure - assuming that the other 95% were warranted and WOULD HAVE BEEN TKRs, the cost savings would be huge. On the other hand, as some of the introduction suggests, chondroplasty may NOT be warranted and might be overused. If NONE of the chondroplasties were warranted (that is to say, they prevented zero of TKAs and had no impact on pain and functioning) then the intervention is a sham. While the authors are clear that their intent is a descriptive study, they also suggest that the information might be helpful for shared decision making. I don't see that argument because, to inform decisions, information on outcomes has to be part of the equation....fundamentally, the authors don't have that. So, two modest revision requests are:  1. To articulate both points above in the discussion: to wit, that chondroplasty might work nearly perfectly (95%) or not at all (essentially 0%), and we simply cannot know that until an RCT is done. 2. Outcomes of the RCT required to make the information presented in the article useful are to understand a) whether chondroplasty is appropriately prescribed b) whether equivalent 'disease states' progress at the same rate with and without chondroplasty and c) whether, in laboratory settings, over a long time period, chondroplasty essentially reduces the demand for TKA. I think making those points - the first on this study's limitations and the second on what would be required to address those limitations - explicit would indeed inform the broader community.
---

VERSION 2 – AUTHOR RESPONSE

Reviewer: 1

Reviewer Name: Feng-Chih Kuo

Institution and Country: Kaohsiung Chang Gung Memorial Hospital, Kaohsiung, Taiwan

Please state any competing interests or state 'None declared': None declared

Please leave your comments for the authors below

The authors did a thorough response to the revised manuscripts. I have no more comments on it. I agree with it being published in BMJ Open.

Thank you

Reviewer: 2

Reviewer Name: William B Weeks, MD, PhD, MBA

Institution and Country: Microsoft, USA

Please state any competing interests or state 'None declared': None declared

Please leave your comments for the authors below

Thanks to the authors for addressing my concerns - primarily that demographics of the comparisons group (TKA with and without prior chondroplasty) had not been included in Table 1. Those data suggest to me that - IF CHONDROPLASTY IS WARRANTED - chondroplasty is pretty effective - only about 5% of TKAs had a prior chondroplasty, so if the narrative is that chondroplasty intervention AVOIDS later (more expensive and intrusive) TKA. A 5% failure rate over a decade isn't bad for a relatively minor procedure - assuming that the other 95% were warranted and WOULD HAVE BEEN TKRs, the cost savings would be huge. On the other hand, as some of the introduction suggests, chondroplasty may NOT be warranted and might be overused. If NONE of the chondroplasties were warranted (that is to say, they prevented zero of TKAs and had no impact on pain and functioning) then the intervention is a sham.

While the authors are clear that their intent is a descriptive study, they also suggest that the information might be helpful for shared decision making. I don't see that argument because, to inform decisions, information on outcomes has to be part of the equation....fundamentally, the authors don't have that.

So, two modest revision requests are:

1. To articulate both points above in the discussion: to wit, that chondroplasty might work nearly perfectly (95%) or not at all (essentially 0%), and we simply cannot know that until an RCT is done.

Thank you for this suggestion. We have added this point to the discussion where we feel it is most relevant and appropriate in the strengths and limitations, third paragraph.

2. Outcomes of the RCT required to make the information presented in the article useful are to understand a) whether chondroplasty is appropriately prescribed b) whether equivalent 'disease states' progress at the same rate with and without chondroplasty and c) whether, in laboratory settings, over a long time period, chondroplasty essentially reduces the demand for TKA.

This point has also been added to the third paragraph of the strengths and limitations which discusses the need for and value from an RCT.

I think making those points - the first on this study's limitations and the second on what would be required to address those limitations - explicit would indeed inform the broader community.

Thank you, we agree and have made these changes as proposed.